# Acute febrile illness among outpatients seeking health care in Bangladeshi hospitals prior to the COVID-19 pandemic

Pritimoy Das[1]*, M. Ziaur Rahman[1], Sayera Banu[1], Mahmudur Rahman[1], Mohammod Jobayer Chisti[1], Fahmida Chowdhury[1], Zubair Akhtar[1], Anik Palit[1], Daniel W. Martin[2], Mahabub Ul Anwar[2], Angella Sandra Namwase[2], Pawan Angra[2], Cecilia Y. Kato[2], Carmen J. Ramos[2], Joseph Singleton[2], Jeri Stewart-Juba[2], Nikita Patel[2], Marah Condit[2], Ida H. Chung[2], Renee Galloway[2], Michael Friedman[2], Adam L. Cohen[2]

1 International Centre for Diarrhoeal Disease Research, Bangladesh (icddr,b), Dhaka, Bangladesh,
2 Centers for Disease Control and Prevention (CDC), Atlanta, Georgia, United States of America

* pritimoydas@gmail.com

**Data Availability Statement:** Data cannot be made publicly available because this Human Subject Research dataset contains potentially sensitive information and hence, are confidential in ethical

## Abstract

Understanding the distribution of pathogens causing acute febrile illness (AFI) is important for clinical management of patients in resource-poor settings. We evaluated the proportion of AFI caused by specific pathogens among outpatients in Bangladesh. During May 2019-March 2020, physicians screened patients aged ≥2 years in outpatient departments of four tertiary level public hospitals. We randomly enrolled patients having measured fever (≥100.4˚F) during assessment with onset within the past 14 days. Blood and urine samples were tested at icddr,b through rapid diagnostic tests, bacterial culture, and polymerase chain reaction (PCR). Acute and convalescent samples were sent to the Centers for Disease Control and Prevention (USA) for *Rickettsia* and *Orientia* (*R/O*) and *Leptospira* tests. Among 690 patients, 69 (10%) had enteric fever (*Salmonella enterica* serotype Typhi or *Salmonella enterica* serotype Paratyphi), 51 (7.4%) *Escherichia coli*, and 28 (4.1%) dengue detected. Of the 441 patients tested for *R/O*, 39 (8.8%) had rickettsioses. We found 7 (2%) *Leptospira* cases among the 403 AFI patients tested. Nine patients (1%) were hospitalized, and none died. The highest proportion of enteric fever (15%, 36/231) and rickettsioses (14%, 25/182) was in Rajshahi. Dhaka had the most dengue cases (68%, 19/28). *R/O* affected older children and young adults (IQR 8–23 years) and was detected more frequently in the 21–25 years age-group (17%, 12/70). *R/O* was more likely to be found in patients in Rajshahi region than in Sylhet (aOR 2.49, 95% CI 0.85–7.32) between July and December (aOR 2.01, 1.01–5.23), and who had a history of recent animal entry inside their house than not (aOR 2.0, 0.93–4.3). Gram-negative Enterobacteriaceae were the most common bacterial infections, and dengue was the most common viral infection among AFI patients in Bangladeshi hospitals, though there was geographic variability. These results can help guide empiric outpatient AFI management.

perspective. icddr,b recognizes the public health, social and intellectual value of providing access to its knowledge data. Data will be provided to interested researchers (Recipients) for upon approval of a Data Licensing Application & Agreement (DLAA) by the icddr,b Data Centre Committee (DCC). Request for icddr,b research data should be addressed to Ms. Armana Ahmed, Head, Research Administration at aahmed@icddrb. org.

**Funding:** This research protocol was funded by the Centers for Disease Control and Prevention (USA) under the cooperative agreement no. GH002259. The funders had no role in study design, data collection and analysis, decision to publish, or preparation of the manuscript.

**Competing interests:** The authors have declared that no competing interests exist.

## Introduction

Acute febrile illness (AFI) is commonly observed in patients suffering from infectious diseases [1]. Globally, AFI is a common cause of outpatient visits and hospital admission; it contributes to considerable morbidity and death among children [2]. AFI burden in adults is also high; adults with febrile illness requiring hospitalization showed 5% to 18% mortality across different settings [3–5]. As a wide spectrum of infectious agents (bacteria, viruses, protozoa) causes AFI, it is important to know about specific pathogens associated with each AFI episode to address the public health challenge of AFI morbidity and mortality [6–8].

Though fever is a common symptom, up to 80% of the fever cases with a shorter duration (<21 days) may remain undiagnosed [9]. A cohort study in Asian countries including Indonesia, Malaysia, Philippines, Thailand, and Vietnam reported that, among 289 participants who had acute fever, the overall incidence density of chikungunya per 100 person-years was 35, *S*. Typhi was 29.4, and dengue was 23.9. Other causes of AFI were *Rickettsia*, leptospirosis, hepatitis A and influenza A [10]. Other studies in countries geographically similar to Bangladesh have reported additional pathogens. Pakistan reported dengue, malaria and Brucella abortus [11, 12], while India reported scrub typhus [13] and dengue [14, 15]. Enteric fever, chikungunya, malaria, leptospirosis, respiratory tract infection, urinary tract infection, and scrub-typhus were found as other aetiologies of AFI in countries of close proximity to Bangladesh [14, 15]. Scrub typhus and enteric fever were also reported as the most common causes of acute fever in Nepal [16–19].

In Bangladesh, a study conducted at Chittagong district identified 9% of AFI cases had undifferentiated febrile illness with 3.4% mortality (4.6% in children and 2.1% in adults) [20]. Labib et al. (2017), tested multiple pathogens among hospital-based febrile patients from December 2008 to November 2009 in Bangladesh [5]. They reported *Rickettsia* (37%) and dengue (9.6%) as the predominant causes of AFI in their study, along with very few cases of *Coxiella*, *Leptospira*, *Bartonella*, and chikungunya virus infections. Other than malaria in endemic zones, most single pathogen studies of acute febrile illness investigated enteric fever [21–28] and dengue [29–37] as these are common diseases in Bangladesh.

Due to the lack of rapid diagnostic capacity, patients suffering from AFI, particularly early in the clinical course when no symptoms can distinguish different aetiologies, pose challenges to their physicians at out-patient departments of any hospital. Additionally, conducting AFI etiologic investigations globally lacks a standardized approach [6]. Clinically, the different causes of acute febrile illnesses may be indistinguishable, and the choice of empiric antibiotics is determined by the etiologic profile, which varies by time, place, and personal factors. To fill the gaps, there is a need for a broad diagnostic testing approach. Recently Hopkins et al. (2020) has taken an initiative through The Febrile Illness Evaluation in a Broad Range of Endemicities (FIEBRE) study to help address these information gaps [38]. FIEBRE was intended to explore AFI in paediatric and adult outpatients and inpatients, using standardised clinical, reference laboratory and social science protocols, in low-resource regions from five sites in sub-Saharan Africa and South-eastern and Southern Asia. There is an absence of published studies investigating the causes of AFI in Bangladesh. Therefore, we conducted an active, prospective surveillance to determine the prevalence and epidemiology of the causative pathogens essential for developing clinical guidelines, diagnostic algorithms, selection of antimicrobials, and management of AFI patients in the outpatient departments of select hospitals in Bangladesh.

## Methods

### Setting

International Centre for Diarrhoeal Diseases Research, Bangladesh (icddr,b), in collaboration with United States Centers for Disease Control and Prevention (U.S. CDC), conducted a

sentinel surveillance for acute febrile illness (AFI) in the outpatient departments (OPDs) of four public hospitals from May 2019 to March 2020. These hospitals are geographically located throughout the country where patients seek care for febrile illness and other health issues (Fig 1). The AFI surveillance sites were three tertiary level government medical college hospitals (Sir Salimullah Medical College & Mitford Hospital, Dhaka; M.A.G. Osmani Medical College hospital, Sylhet; and Rajshahi Medical College Hospital, Rajshahi) and one secondary level government district hospital (Sadar Hospital, Feni).

## Patient enrolment

We assigned two trained personnel (one physician and one medical technologist) to each hospital to screen all patients attending the medicine and pediatric outpatient departments for suspected cases of acute febrile illness. Patients aged $\geq 2$ years with a history of reported fever within the previous 14 days of outpatient visit were identified, subject to the following exclusion criteria: symptoms of a focused infection like cellulitis, abscess, boil, or local skin infection; a history of trauma; follow-up cases of known cause of fever including a diagnosed case of tuberculosis; post-operative cases presenting with fever within 30 days; and taking any antibiotics in the past 24 hours. Using digital clinical thermometers, the physician measured the suspected AFI patients' oral and/or axillary temperatures (whichever was possible) and recorded the highest temperature obtained. Eligible patients had a measured fever (oral/axillary temperature $\geq 100.4°F$). The staff kept two separate logbooks with serial numbers of eligible patients for the adult and pediatric groups on each enrolment day. Staff offered some of the eligible patients to participate in AFI surveillance each day using a random sampling technique, obtained written informed consent, and then enrolled them in our surveillance. Fig 2 shows the detailed enrolment process.

## Data collection

Staff used a structured questionnaire to collect data on socio-demographics, travel history, animal exposure, and clinical characteristics. Field personnel used tablet computers to collect data, which they synchronized with an icddr,b server via mobile internet. This system enabled the research team to centrally monitor the enrolment across all hospitals in real-time from Dhaka. After 30 days, the physician followed up with each enrolled patient via mobile phone calls or face-to-face interviews to register the outcome of their illnesses and update the database accordingly.

## Specimen collection

On the day of enrolment, field staff collected blood and urine specimens from each enrolled patient using standard aseptic and clean catch techniques. During the follow-up visit, after 30 days, a convalescent blood specimen was collected to test for *Rickettsia* and *Leptospira*.

## Testing specimen for AFI pathogens

**At enrolment in hospital.** Staff conducted rapid diagnostic tests, as per manufacturer's instructions, on patients' blood for dengue (NS1), chikungunya (IgM and IgG), malaria (*m. falciparum* and *m. vivax*) and Zika virus (IgM and IgG) (RDT kits manufactured by Biopanda reagents, Belfast, UK, 'S1 Table. List of pathogens and tests used for AFI surveillance, Bangladesh') and reported the result to the treating physician immediately.

**At icddr,b laboratory.** Blood specimens for culture (collected in Bactec bottles) and urine specimens for culture were transported to the icddr,b laboratory at room temperature by staff

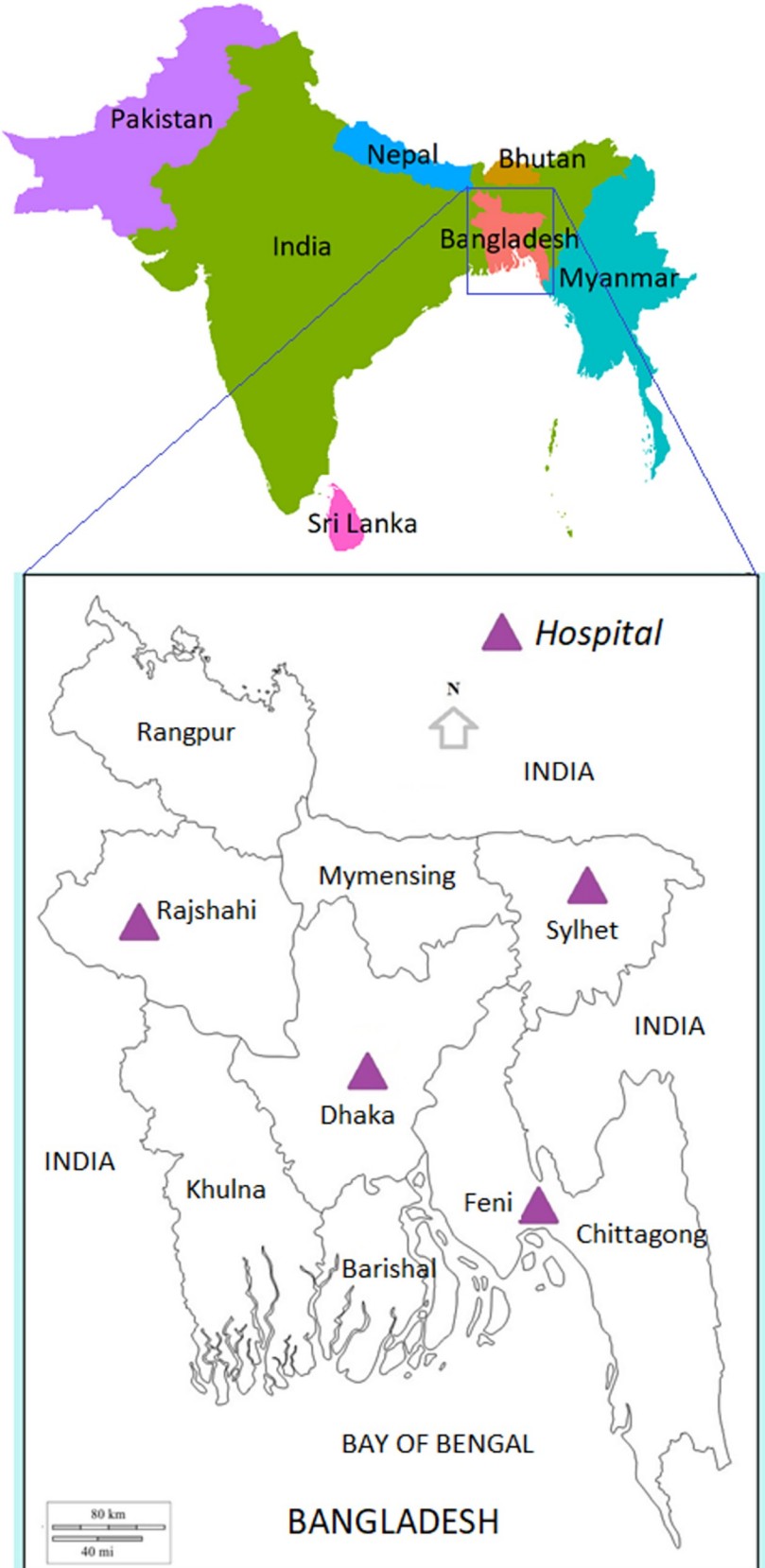

**Fig 1. Bangladesh map showing hospital locations for acute febrile illness (AFI) surveillance in Bangladesh.**

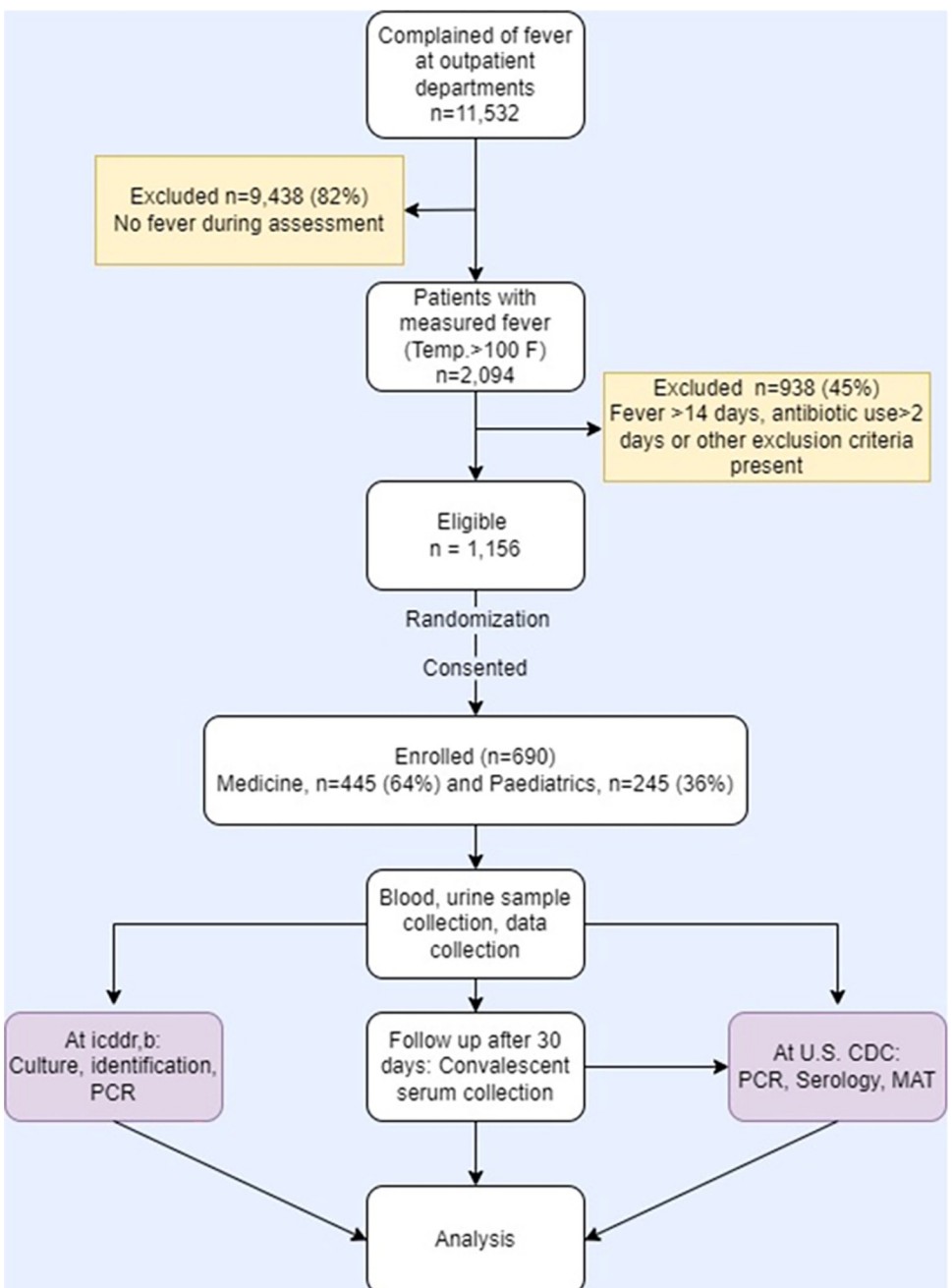

**Fig 2. Flow diagram of surveillance enrollments among adults and children who visited outpatient departments for acute febrile illness (AFI) in Bangladesh, May 2019-March 2020.**

or courier service. Blood and serum samples for polymerase chain reaction (PCR) analysis were transported in a cooler box at 2–8°C. All specimens were transported to the laboratory within 24 hours of collection. Positive blood and urine cultures were subjected to bacterial identification and antibiotic susceptibility testing in a fully automated VITEK 2 system [39]. We tested for a number of pathogens using different VITEK cards [40–42]. In the Virology Laboratory at icddr,b, acute blood samples were tested for a panel of pathogens; *Rickettsia* spp., *Orientia tsutsugamushi*, *Leptospira* spp., *Brucella* spp., *Coxiella burnetti*, Crimean-Congo

haemorrhagic fever (CCHF), Hepatitis E virus, West Nile virus, and Zika virus by real-time RT-PCR assays with the technical support from the U.S. CDC. The primers and probes specific for the targeted pathogens have already been published elsewhere [43]. The total nucleic acid from blood samples was extracted using a chemagic Viral NA/gDNA 200 Kit H96 (PerkinElmer, MA, USA) in a chemagic™ 360 instrument (PerkinElmer chemagen Technologie GmbH). The one-step real-time RT-PCR assays were conducted using the iTaq universal probes one-step kit (Bio-Rad Laboratories, CA, USA) in a Bio-Rad CFX96 Touch real-time PCR system.

**At U.S. CDC laboratory.** We tested patients for rickettsioses and leptospirosis who provided both acute and convalescent serum samples. At the Rickettsial Zoonoses Branch laboratory, *Rickettsia* spp. and *Orientia tsutsugamushi* real-time PCR and reverse transcriptase real-time PCR tests to detect DNA and RNA/DNA respectively were performed on acute patient blood. Spotted fever group *Rickettsia*, typhus group *Rickettsia*, and *Orientia tsutsugamushi* Immunoglobulin G (IgG) indirect immunofluorescence antibody (IFA) assays were used to identify patient immune response to these bacteria in paired acute and convalescent serum. This testing was an additional method to detect rickettsioses and scrub typhus infections. To be confirmed positive by serology, both 1) an IgG convalescent titer of 1:64 or greater AND 2) a 4-fold or greater rise in titer from the acute to convalescent was required. Leptospirosis diagnosis was conducted in the Zoonoses and Select Agent Laboratory at U.S. CDC using the microscopic agglutination test (MAT). Briefly, live antigens representing different serogroups undergo reaction with patient serum samples to detect agglutinating antibodies; detailed method described elsewhere[44, 45].

### Data analysis

The data management and analyses were performed using the software Stata v.14 (Stata Corp LP, College Station, TX, USA). We summarized all categorical variables using frequencies and percentages. We performed univariable and multivariable logistic regression analysis to interpret any association between explanatory and outcome variables. For this analysis, we used four distinct binary outcome variables: diagnosed enteric fever, *E. coli* urinary tract infection, dengue, and rickettsioses. We used patients' age, sex, urban-rural residential status, occupation, temperature, geographic location of the hospitals, history of animal entry inside the house as the explanatory variables. Because of sparse data, we used a penalized maximum likelihood estimation method for reducing bias in generalized regression models [46–49]. The univariable analysis was used to identify factors associated with the outcomes. A p-value less than 0.05 was considered statistically significant. Significant independent variables, in the univariate analysis, were controlled for in the multivariable analysis.

### Ethical consideration

The protocol was reviewed and approved by the institutional review boards (IRB; Research Review Committee and Ethical Review Committee, number PR-18071) of the International Centre for Diarrhoeal Disease Research, Bangladesh (icddr,b).We obtained written, informed consent from the patients or caregivers before enrolment.

### Results

#### Description and etiology of AFI cases

From May 2019 to March 2020, we enrolled 690 patients with acute febrile illness (AFI) from four different hospitals' outpatient departments. The enrolled patients' median age was 17 years (IQR: 8–25 years), and 435 (63%) were male. Approximately two-thirds of all AFI

**Table 1. Demographic characteristics of enrolled acute febrile illness (AFI) patients in Bangladesh, May 2019-March 2020.**

| | Sylhet (N = 150) n (%) | Dhaka (N = 233) n (%) | Rajshahi (N = 231) n (%) | Feni (N = 75) n (%) | Total (N = 690) n (%) |
|---|---|---|---|---|---|
| **Age in years** | | | | | |
| Median (IQR) | 17 [8, 25] | 18 [11, 30] | 17 [9, 22] | 15 [6, 24] | 12 [6, 25] |
| **Age groups** | | | | | |
| [0–5] | 10 (4%) | 18 (8%) | 11 (15%) | 22 (15%) | 61 (9%) |
| [5–10] | 39 (17%) | 40 (17%) | 18 (24%) | 38 (25%) | 136 (20%) |
| [10–15] | 44 (19%) | 31 (13%) | 8 (11%) | 18 (12%) | 101 (15%) |
| [15–20] | 29 (12%) | 51 (22%) | 12 (16%) | 15 (10%) | 107 (16%) |
| [20–25] | 26 (11%) | 46 (20%) | 8 (11%) | 17 (11%) | 97 (14%) |
| [25–30] | 26 (11%) | 11 (5%) | 10 (13%) | 16 (11%) | 63 (9%) |
| [30–35] | 20 (9%) | 11 (5%) | 3 (4%) | 7 (5%) | 41 (6%) |
| [35–40] | 10 (4%) | 9 (4%) | 1 (1%) | 4 (3%) | 24 (3%) |
| [40–45] | 12 (5%) | 7 (3%) | 3 (4%) | 4 (3%) | 26 (4%) |
| [45+] | 17 (7%) | 7 (3%) | 1 (1%) | 9 (6%) | 34 (5%) |
| **Sex** | | | | | |
| Male | 435 (63%) | 147 (63%) | 147 (64%) | 48 (64%) | 93 (62%) |
| Female | 255 (37%) | 86 (37%) | 84 (36%) | 27 (36%) | 57 (38%) |
| **Urban/rural** | | | | | |
| Urban | 471 (68%) | 230 (99%) | 136 (59%) | 37 (49%) | 67 (45%) |
| Rural | 219 (32%) | 3 (1%) | 95 (41%) | 38 (51%) | 83 (55%) |
| **Schooling years** | | | | | |
| [0] | 96 (14%) | 16 (7%) | 30 (13%) | 18 (24%) | 32 (21%) |
| [1–5] | 241 (35%) | 103 (44%) | 64 (28%) | 24 (32%) | 49 (33%) |
| [6–10] | 152 (22%) | 80 (34%) | 43 (19%) | 11 (15%) | 18 (12%) |
| [11–12] | 142 (21%) | 25 (11%) | 81 (35%) | 10 (13%) | 26 (17%) |
| [>12] | 59 (9%) | 9 (4%) | 13 (6%) | 12 (16%) | 25 (17%) |
| **Occupation** | | | | | |
| Unemployed | 194 (28%) | 54 (23%) | 51 (22%) | 25 (33%) | 64 (43%) |
| Job holder | 49 (7%) | 21 (9%) | 7 (3%) | 3 (4%) | 18 (12%) |
| Business | 33 (5%) | 24 (10%) | 3 (1%) | 3 (4%) | 3 (2%) |
| Student | 342 (50%) | 101 (43%) | 154 (67%) | 36 (48%) | 50 (33%) |
| Other* | 72 (10%) | 33 (14%) | 16 (7%) | 8 (11%) | 15 (10%) |

*Farmer, day-labour, small shop owner, rickshaw/van puller, driver etc.

patients (471, 68%) reported living in an urban setting. However, just over half of patients in Sylhet (83, 55%) and Feni (38, 51%) reported living in a rural setting. Only a few patients (59, 9%) had a level of education >12 years, (Table 1).

Acute blood was collected from all 690 enrolled patients, but we could not collect/test urine from 9 (1.3%) patients due to technical problems (For example, a 2-year-old female child was unable to provide urine in outpatient settings, or a delayed (>24 hour) arrival of urine from Rajshahi hospital, a facility far from the Dhaka icddr,b laboratory) and convalescent serum from 249 (36%) patients after one month of enrolment due to loss to follow up. During the surveillance period, pathogens were detected from 182 (26%) of 690 AFI patients. *Salmonella enterica* (bacteria causing enteric fever) was the most common pathogen identified in patients (10%, 69), (Table 2). Rickettsial diseases including scrub typhus were found in 8.8% of the AFI patients tested for these diseases (39/441). Urinary tract infection caused by *Escherichia coli* was the third most common cause of AFI (7.5%, 51 of 681 tested). Dengue was identified in

**Table 2. Distribution of pathogens identified among acute febrile illness (AFI) patients in Bangladesh, May 2019-March 2020.**

| Disease/Pathogen | Test | Timing of Specimen | # positive/# tested | Percentage |
|---|---|---|---|---|
| **Enteric fever (Typhoid)** | **Blood culture** | **Acute** | **69/690** | **10.0%** |
| *S.* Typhi | Blood culture | Acute | 49/690 | 7.1% |
| *S.* Paratyphi | Blood culture | Acute | 20/690 | 2.9% |
| **Rickettsia** | **Positive by any either PCR* or serology** | **Acute and Convalescent** | **39/441** | **8.8%** |
| *Orientia tsutsugamushi* | PCR | Acute | 21/441 | 4.8% |
| *Other Rickettsia spp.* | PCR | Acute | 6/441 | 1.4% |
| *Orientia tsutsugamusi* | Serology | Acute and Convalescent: Seroconversion | 17/441 | 3.8% |
| Typhus group *Rickettsia* | Serology | Acute and Convalescent: Seroconversion | 3/441 | 0.7% |
| Spotted fever group *Rickettsia* | Serology | Acute and Convalescent: Seroconversion | 1/441 | 0.2% |
| ***E. coli*** | **Urine culture** | **Acute** | **51/681** | **7.5%** |
| **Dengue** | **RDT NS1** | **Acute** | **28/690** | **4.1%** |
| **Leptospirosis** | **MAT** | **Acute** | **7/403** | **1.7%** |
| *L. interrogans* serovar Bratislava | MAT | Acute and convalescent | 4/403 | 1.0% |
| *L. interrogans* serovar Canicola | MAT | Acute and convalescent | 1/403 | 0.2% |
| *L. interrogans* serovar Mankarso | MAT | Acute and convalescent | 1/403 | 0.2% |
| *L. borgpetersenii* serovar Tarrasovi | MAT | Acute and convalescent | 1/403 | % |

*Note: PCR = Polymerase chain reaction. RDT = Rapid diagnostic tests, MAT = Microscopic agglutination test. Some patients were both PCR and sero-positive for Rickettsia: Among the 27 PCR positive cases, 9 (33%) patients were positive by both PCR and seroconversion.

4.1% (28/690) of cases, followed by leptospirosis 2% (7/403). Of the tests for other viruses, Hepatitis E was identified in 1 (0.14%) patient.

**Enteric fever (*Salmonella enterica* serotype Typhi and Paratyphi).** *S.* Typhi or Paratyphi were found in people of all ages ranging from 2 to 45 years old, with the highest percentage in the 15-19-year age group (20%, 21/107). It was more common in Rajshahi (15.5%) followed by urban Dhaka (11.2%). Multivariable logistic regression model showed that this infection was more likely to occur among male patients than female (aOR 2.13; 95% CI: 1.15–3.92), among patients living in an urban than a rural residence (aOR 2.59; 95% CI: 1.18–5.7), in Rajshahi (aOR 2.77; 95% CI: 1.05–7.33) and Dhaka (aOR 1.7; 95% CI: 0.62–4.65) than Sylhet region, and in patients presenting with a high fever (>103˚F) compared to patients with a lower temperature (<102˚F), (aOR 3.09; 95% CI: 1.25–7.62, Table 3A).

Among the AFI aetiologies, pathogens causing enteric fever were found throughout the calendar year. The proportion, of detected enteric fever cases among the samples tested per month, was the lowest (3.7%) in November, then gradually increased in each month and peaked in March (22.9%). Between May and October, the average monthly typhoid detection rate was 7.3%. Another peak of enteric fever was noted in September (14.3%), (Fig 3).

**Dengue.** Dengue virus was detected in almost all ages. This viral infection was mainly detected in Dhaka: 8% of all samples from Dhaka alone were dengue positive. Dengue was found in all three other locations (frequency varied from 1–3%). Of all dengue cases detected from this AFI surveillance, 19 (68%) were detected from Dhaka site alone. Regression analysis suggested significantly higher dengue risk in Dhaka region compared to Rajshahi (OR 7.5; 95% CI: 1.8–31.2). Patients with a high temperature (>103˚F) were approximately seven times more likely to have dengue infection and, patients with a mid-range temperature (102–103˚F) were approximately three times more likely to suffer from dengue, compared to patients with mild fever (aOR 6.9; 95% CI: 2.4–20 and aOR 2.83; 95% CI: 1.15–6.9 respectively, (Table 3B).

**Table 3. Univariate and multivariate regression models for pathogens commonly detected in acute febrile illness (AFI) surveillance in Bangladesh, May 2019-March 2020.**

| a. Enteric fever | | | | |
|---|---|---|---|---|
| **Factors** | | **Enteric fever** | | **OR (univariable)** | **OR (multivariable)** |
| | | **No** | **Yes** | **(95% CI, p-value)** | **(95% CI, p-value)** |
| Age (year) | [0–5] | 59 (96.7) | 2 (3.3) | Ref | Ref |
| | [5–10] | 130 (95.6) | 6 (4.4) | 1.19 (0.27–5.26, p = 0.823) | 0.72 (0.14–3.75, p = 0.694) |
| | [10–15] | 85 (84.1) | 16 (15.8) | **4.59 (1.17–18.09, p = 0.029)** | 2.25 (0.47–10.87, p = 0.313) |
| | [15–20] | 86 (80.4) | 21 (19.6) | **5.92 (1.53–22.83, p = 0.01)** | 2.73 (0.61–12.25, p = 0.191) |
| | [20–25] | 84 (86.6) | 13 (13.4) | **3.8 (0.95–15.26, p = 0.06)** | 1.94 (0.44–8.55, p = 0.383) |
| | [25–30] | 56 (88.9) | 7 (11.1) | 3.16 (0.72–13.84, p = 0.127) | 2.18 (0.47–10.02, p = 0.318) |
| | [30–35] | 40 (97.6) | 1 (2.4) | 0.88 (0.11–6.94, p = 0.905) | 0.52 (0.06–4.25, p = 0.544) |
| | [35–40] | 22 (91.7) | 2 (8.3) | 2.64 (0.43–16.3, p = 0.295) | 2.19 (0.34–13.95, p = 0.406) |
| | [40–45] | 25 (96.2) | 1 (3.8) | 1.4 (0.18–11.17, p = 0.751) | 1.09 (0.13–9.05, p = 0.933) |
| Sex | Female | 240 (94.1) | 15 (5.9) | Ref | Ref |
| | Male | 383 (88.0) | 52 (12.0) | **2.21 (1.23–3.9, p = 0.008)** | **2.13 (1.15–3.92, p = 0.015)** |
| Residence | Rural | 210 (95.9) | 9 (4.1) | Ref | Ref |
| | Urban | 411 (87.3) | 60 (12.7) | **3.25 (1.61–6.5, p = 0.001)** | **2.59 (1.18–5.69, p = 0.018)** |
| Location | Sylhet | 146 (96.7) | 5 (3.3) | Ref | Ref |
| | Rajshahi | 195 (84.4) | 36 (15.5) | **4.97 (1.98–12.5, p = 0.001)** | **2.77 (1.05–7.33, p = 0.04)** |
| | Dhaka | 207 (88.8) | 26 (11.2) | **3.4 (1.33–8.73, p = 0.011)** | 1.7 (0.62–4.65, p = 0.301) |
| | Feni | 73 (97.3) | 2 (2.7) | 0.91 (0.2–4.15, p = 0.899) | 0.61 (0.13–2.87, p = 0.529) |
| Fever | Low <102˚F | 451 (91.1) | 44 (8.9) | Ref | Ref |
| | Medium (102–103˚F) | 138 (89.0) | 17 (10.9) | 1.28 (0.71–2.3, p = 0.405) | 1.07 (0.57–1.98, p = 0.838) |
| | High >103˚F | 32 (80.0) | 8 (20.0) | **2.65 (1.17–6, p = 0.019)** | **3.09 (1.25–7.62, p = 0.014)** |
| Student | No | 328 (94.3) | 20 (5.7) | Ref | Ref |
| | Yes | 293 (85.7) | 49 (14.3) | **2.7 (1.6–4.62, p = 0.001)** | 1.52 (0.7–3.29, p = 0.288) |
| Time | Jan-June | 236 (87.0) | 35 (13.0) | Ref | Ref |
| | July-Dec | 385 (91.9) | 34 (8.1) | **0.55 (0.34–0.92, p = 0.022)** | 0.65 (0.38–1.1, p = 0.11) |

| b. Dengue fever | | | | |
|---|---|---|---|---|
| **Factors** | | **Dengue** | | **OR (univariable)** | **OR (multivariable)** |
| | | **No** | **Yes** | **(95% CI, p-value)** | **(95% CI, p-value)** |
| Residence | Rural | 215 (98.2) | 4 (1.8) | Ref | Ref |
| | Urban | 447 (94.9) | 24 (5.1) | 2.6 (0.94–7.2, p = 0.064) | 1.38 (0.37–5.12, p = 0.625) |
| Location | Rajshahi | 229 (99.1) | 2 (0.9) | Ref | Ref |
| | Dhaka | 214 (91.8) | 19 (8.2) | **8.34 (2.20–31.54, p = 0.002)** | **7.57 (1.83–31.29, p = 0.005)** |
| | Sylhet | 146 (96.7) | 5 (3.3) | 3.44 (0.76–15.59, p = 0.108) | 3.86 (0.83–17.93, p = 0.084) |
| | Feni | 73 (97.3) | 2 (2.7) | 3.12 (0.53–18.37, p = 0.208) | 3.94 (0.63–24.63, p = 0.142) |
| Fever | Low <102˚F | 483 (97.6) | 12 (2.4) | Ref | Ref |
| | Med (102–103˚F) | 146 (94.2) | 9 (5.8) | **2.50 (1.05–5.94, p = 0.037)** | **2.83 (1.15–6.99, p = 0.023)** |
| | High >103˚F | 33 (82.5) | 7 (17.5) | **8.65 (3.28–22.85, p<0.001)** | **6.96 (2.42–20.01, p<0.001)** |
| Time | Jan-June | 269 (99.6) | 1 (0.4) | Ref | Ref |
| | July-Dec | 393 (93.6) | 27 (6.4) | **12.55 (2.40–65.42, p = 0.003)** | **13.75 (2.59–72.93 p = 0.002)** |

| c. Fever from *Escherichia coli* urinary tract infection | | | | |
|---|---|---|---|---|
| | | ***E. Coli*** | | **OR (univariable)** | **OR (multivariable)** |
| **Factors** | | **No** | **Yes** | **(95% CI, p-value)** | **(95% CI, p-value)** |
| Age in years | 0–35 | 566 (93.4) | 40 (6.6) | Ref | Ref |
| | 35+ | 73 (86.9) | 11 (13.1) | **2.18 (1.08–4.40, p = 0.028)** | 1.37 (0.63–3.01, p = 0.43) |
| Sex | Female | 221 (86.7) | 34 (13.3) | Ref | Ref |

(*Continued*)

**Table 3.** (Continued)

| | | No | Yes | OR (univariable) (95% CI, p-value) | OR (multivariable) (95% CI, p-value) |
|---|---|---|---|---|---|
| | Male | 418 (96.1) | 17 (3.9) | **0.26 (0.14–0.48, p<0.001)** | **0.26 (0.12–0.53, p<0.001)** |
| Location | Rajshahi | 226 (97.8) | 5 (2.2) | Ref | Ref |
| | Dhaka | 210 (90.1) | 23 (9.9) | **4.59 (1.78–11.85, p = 0.002)** | **4.17 (1.59–10.91, p = 0.004)** |
| | Sylhet | 136 (90.1) | 15 (9.9) | **4.67 (1.72–12.65, p = 0.002)** | **3.95 (1.41–11.06, p = 0.009)** |
| | Feni | 67 (89.3) | 8 (10.7) | **5.18 (1.71–15.68, p = 0.004)** | **4.95 (1.59–15.37, p = 0.006)** |
| Student | No | 312 (89.7) | 36 (10.3) | Ref | Ref |
| | Yes | 327 (95.6) | 15 (4.4) | **0.40 (0.22–0.74, p = 0.004)** | 0.55 (0.26–1.15, p = 0.112) |
| Housewives | No | 561 (94.0) | 36 (6.0) | Ref | Ref |
| | Yes | 78 (83.9) | 15 (16.1) | **3.03 (1.60–5.75, p = 0.001)** | 0.91 (0.38–2.18, p = 0.839) |

| | | | | | |
|---|---|---|---|---|---|
| **d. Rickettsial fever** | | | | | |
| **Factors** | | *Rickettsia* | | OR (univariable) | OR (multivariable) |
| | | **No** | **Yes** | **(95% CI, p-value)** | **(95% CI, p-value)** |
| Location | Sylhet | 84 (95.45) | 4 (4.55) | Ref | Ref |
| | Rajshahi | 157 (86.07) | 25 (13.74) | **3.04 (1.08–8.57, p = 0.035)** | 2.49 (0.85–7.32, p = 0.096) |
| | Dhaka | 128 (94.8) | 7 (5.19) | 1.09 (0.32–3.64, p = 0.881) | 1.17 (0.35–3.93, p = 0.789) |
| | Feni | 33 (91.67) | 3 (8.33) | 1.96 (0.46–8.39, p = 0.363) | 1.26 (0.27–5.93, p = 0.771) |
| Animal entry | No | 317 (92.96) | 23 (6.74) | Ref | Ref |
| | Yes | 84 (84.00) | 16 (16.0) | **2.64 (1.34–5.19, p = 0.005)** | 2.00 (0.93–4.30, p = 0.07) |
| Time | Jan-June | 141 (95.2) | 7 (4.8) | Ref | Ref |
| | July-Dec | 260 (88.7) | 32 (10.9) | **2.34 (1.03–5.32, p = 0.042)** | **2.3 (1.01–5.2 p = 0.04)** |

Note: Bold = Significant association

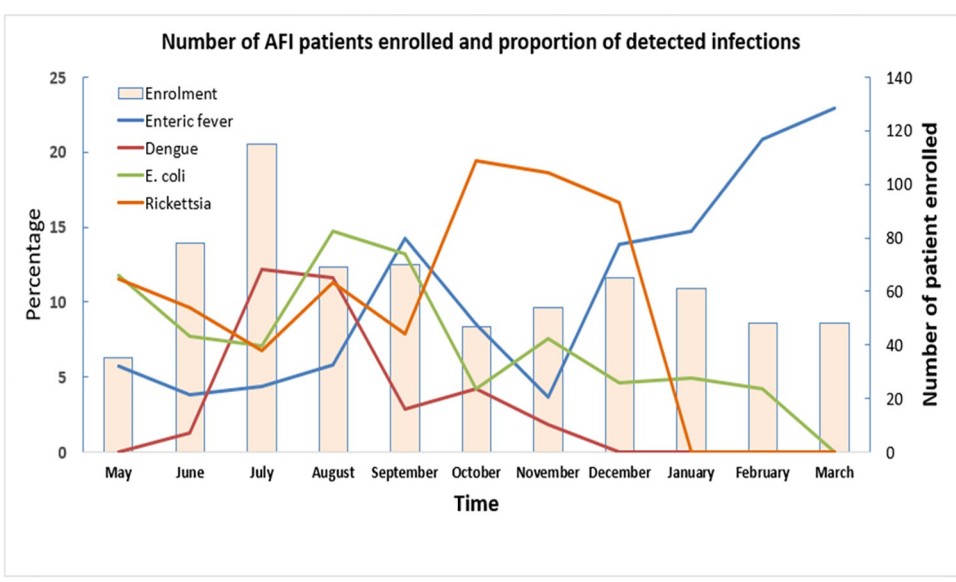

*NB. Denominator for a pathogen was the number of patients tested for that pathogen in a particular month, thereby, the denominator changed depending on the pathogens tested.*

**Fig 3. Proportionate distribution of the most common infections over time among the acute febrile illness (AFI) patients in Bangladesh, May 2019-March 2020.**

We found the first dengue patient in June, then the proportion of dengue among patients enrolled each month raised to the peak during July-August season (12%). Dengue was detected mostly in the second half of the calendar year, rather than from January to June (aOR 13.7; 95% CI: 2.6–72.9). We did not detect any dengue during December to March in Bangladesh (Fig 3).

**Urinary tract infection (UTI) caused by *Escherichia coli*.** *Escherichia coli* UTI was more common in the extreme age groups, especially among children under the age of five (15%), and the 45–50 years of age group (18%). Geographically, *E. coli* was detected less frequently in Rajshahi region (2%) and most frequently in Feni (11%). Compared to patients suffering from acute fever in Rajshahi, patients from Dhaka (aOR 4.2; 95% CI: 1.6–10.9), Sylhet (aOR 3.9; 95% CI: 1.4–11.0) and Feni (aOR 4.9; 95% CI: 1.6–15.4) were more likely to be infected with *E. coli*. Male AFI patients were less likely to suffer from this infection than females (aOR 0.26; 95% CI: 0.12–0.53), (Table 3C). The highest proportion of *E. coli* infections were found in August (14.5%). Detection rate then gradually decreased month by month, with a slight upward trend in November (7.4%) and May. (11.4%), (Fig 3).

**Rickettsiosis.** We could not test 249 (36%) patients for rickettsiosis because they did not provide convalescent samples. Of the 441 tested, a total of 39 (8.8%) patients were acutely infected with at least one rickettsial pathogen detected by either PCR, seroconversion, or both. *O. tsutsugamushi* or *Rickettsia spp.* were detected in 27 (6.1%) patients using DNA and RNA detection by real-time PCR. Seroconversions for *O. tsutsugamushi* 3.9% (17/441), typhus group *Rickettsia* 0.7% (3/441), and spotted fever group *Rickettsia* 0.2% (1/441) were observed. Of all patients with serum anti-*Rickettsia* titer ≥1:64, *Orientia tsutsugamushi* was the most common pathogen (140, 32%), followed by spotted fever group *Rickettsia* (31, 7.03%) and typhus group *Rickettsia* (14, 3.2%).

Rickettsioses occurred mostly in young adults (IQR 8–23 years) and were more common in the 21–25 years age-group (17.14%, 12/70). The highest frequency of rickettsiosis was in Rajshahi (13.7%, 25/182). On univariable regression model analysis, rickettsiosis was more likely to occur among patients from Rajshahi region compared to Sylhet (OR 3.04; 95% CI: 1.08–8.57) and was more likely to be found in patients who had a history of recent animal entry inside their house than not (OR 2.6, 95% CI 1.34–5.19). These statistical significances were lost in the multivariable model (Table 3D). Interestingly, of the 28 patients with a documented temperature of >103 degrees, zero had evidence of confirmed rickettsiosis compared with 39/413 (9.4%) with a temperature of 103 degrees of less, although this did not reach statistical significance.

**Leptospirosis.** 403 patients' both acute and convalescent samples were tested for leptospirosis. Utilizing the MAT test, we were able to identify seven patients with recent leptospiral infection (7/441, 1.6%). Most of them exhibited antibodies against *L. interrogans* serovar Bratislava (57%). Other reactive serovars were *L. interrogans* serovar Canicola, *L. interrogans* serovar Mankarso, and *L. borgpetersenii* serovar Tarrasovi (14% each).

## Clinical treatment and outcome

Almost all AFI patients were treated at home by hospital physicians, with the exception of 9 (1.3%) AFI patients who required hospitalization; seven enrolled from medicine OPD and two from paediatrics OPD. Among the admitted AFI patients, age varied from 7 to 50 years; seven were male (78%), four (45%) were from Dhaka, two (22%) each from Rajshahi and Feni, and one (11%) from Sylhet hospital. At the time of admission, four patients (45%) were diagnosed with dengue fever by our AFI rapid diagnostic tests, one was clinically suspected of having typhoid fever, and other four febrile patients were admitted without a specific diagnosis. These

5 hospitalized patients did not end up having a confirmed diagnosis based on the negative results of all of our study laboratory investigations. The length of hospitalization ranged from 4 to 8 days. Only two of the hospitalized AFI patients received antibiotics: a suspected typhoid fever patient was given injectable ceftriaxone, and a pediatric patient was given azithromycin. During our one-month follow-up, all hospitalized patients reported no illness, and none of the 690 AFI patients died.

## Discussion

Findings from our surveillance show *Salmonella enterica*, *Rickettsia*, and urinary *Escherichia coli* as the most common bacterial pathogens among AFI patients. Dengue was the most frequently detected viral infection, predominant in Dhaka. Current investigation also identified a few cases of *Leptospira* and one case of Hepatitis E among the enrolled AFI patients.

In contrast to many studies focusing on a single pathogen [21, 22, 24, 25, 31, 33, 37, 50–63], this surveillance tested for multiple pathogens causing acute febrile illness in Bangladesh. Our findings extended the work of Labib et al (2017), where the researchers tested multiple pathogens among hospital-based febrile patients, both inpatient and outpatient, from December 2008 to November 2009 in Bangladesh [5]. Compared with Labib et al, this current surveillance added value by investigating more samples focusing on outpatient departments, samples were tested irrespective of their presenting sign-symptoms, and we added more robust testing protocols including blood and urine culture, rapid tests, PCR tests and serological tests for acute and convalescent samples. In both investigations, only one hospital was overlapped (Rajshahi).

Enteric fever was the most commonly identified cause of AFI in this surveillance. A multi country study (Bangladesh, Nepal and Pakistan) reported 5.9% confirmed enteric fever patients among suspected enteric fever cases from the outpatient departments [52]. Others identified enteric fever among 3.6% of hospitalized children in Dhaka, of whom 55% were male [24]. Surveillance for Enteric Fever in Asia Project (SEAP) [22], a large scale project recommended to introduce typhoid conjugate vaccine as a preventive tool against enteric fever, collected data in 2017–2019 and reported high burden of hospitalization due to enteric fever, especially among children of <5 years of age in Bangladesh [64]. In contrast to our surveillance, SEAP was limited to children aged <15 years. In our study, we found the overall proportion of confirmed Typhoid (10%) to be higher than these other studies. This could be due to enrollment of patients only with a documented fever which unfortunately is yet to be routinely performed in most busy outpatient departments in Bangladesh. We also found a strong correlation with a higher documented temperature reading (especially >103 degrees) which again points at the potential clinical benefits of documenting temperature in all outpatients with suspected infections. Our study also observed that the proportion of confirmed Typhoid reached a peak (20%) with the 15–20 years age group and was common among patients aged 10–40 years. A multi-country population-based study found that the seroincidence of *Salmonella enterica* serotype Typhi exposure in Bangladesh was highest in 15–29 years age group and individuals aged 5 to 9 years had the highest adjusted incidence of typhoid fever [65]. This could have significant implications for the timing of any Typhoid vaccine introductions in Bangladesh or the region.

*Salmonella* is transmitted through contaminated water and food. Due to cultural norms, males are frequently going outside for routine tasks and more likely consuming food and water away from home than females, which may explain why enteric fever was more prevalent among males. In addition, the reason for an increase in males with enteric fever could be due to differing healthcare utilization practices. Additionally, poor water and sanitation in the

urban settings could be a contributing factor. Improved water and sanitation system, monitoring food serving vendors, as well as public awareness before taking outside food and drinks may be helpful to reduce the disease burden.

The median cost of illness per case of enteric fever from the patient and caregiver perspective and healthcare provider perspective has been found to be US $64.03 and US $58.64 respectively [21]. Qadri et al. (2021) conducted a randomized control vaccine trial for enteric fever and reported that, overall, Vi-tetanus toxoid conjugate vaccine efficacy against typhoid fever was ranging from 80% to 88% for vaccinated children in urban Dhaka [66]. Given that typhoid was more common in urban areas in this surveillance, it is important to use this vaccine efficacy evidence to introduce typhoid vaccine programs in urban areas. Since treating typhoid is costly, vaccination programs will be more economically beneficial [67], for the Ministry of Health, in addressing the public health challenge of large numbers of typhoid cases.

We found that rickettsial pathogens were confirmed as the cause of 9.1% of patients with AFI. In other studies, *Rickettsia* have been identified in 19–48% of the patients presenting at hospitals with acute fever in Bangladesh [5, 63, 68]. However, it must be noted that the definitions used for confirmed or suspected rickettsial infections varies widely from study to study, with many studies using a single elevated antibody titer as their criteria for a case. This is believed to result in an overestimation of the true number of acute rickettsial infections. Given that our study used a very strict case definition, the fact that 8.8% of patients were confirmed to have acute rickettsiosis is both noteworthy and surprisingly high. Its frequency among the AFI outpatients is similar to that of enteric fever and may be important for clinicians. Clinicians should consider testing febrile patients for *Rickettsia* in the differential diagnosis of AFI. CDC guidance is to treat upon suspicion of disease and not to wait for laboratory confirmation for timely treatment[69]. The understanding of rickettsial disease prevalence in Bangladesh could help in the timely administration of appropriate antibiotic treatment [70, 71]. Detailed history of tick bites, lice, and skin examination for eschar marks backed by laboratory testing should help in clinically diagnosing and treating AFI patients appropriately.

Our investigation revealed that urinary tract infection caused by *E. coli* infection was the third most common cause of AFI, which is one of the commonly grown pathogens in urine cultures. Other researchers have found similar results: In Dhaka, *E. coli* was more frequently isolated from the urine samples (6.2% all UTI suspected women, 69.0% of total bacteriuria) [61]. In another study, out of the 551 tested samples, the most prevalent was *E. coli* (98, 17.8%) and majority of the patients (73.3%) were female [72]. *E. coli* was the predominant isolate (59.3%) in Rajshahi [58], Dhaka (69.2%) [73] and in Mymensingh (48.5%) [53]. *E. coli* UTI should be considered in the diagnosis of female AFI patients, especially >35 years of age anywhere in Bangladesh, and urine culture is recommended for appropriate treatment and management.

We also found dengue fever commonly among our AFI patients, mostly in Dhaka district. Data from our surveillance showed a peak of dengue in July-August months, similar to data published in national reports [74]. This could be due to the rainy season in Bangladesh that happens in this time period. The crowding of Dhaka produces conditions favorable to *Aedes* mosquito breeding, which may account for the elevated detection of dengue in the city.

Our study was conducted in the busy outpatient departments of large government hospitals. We expected that most patients would observe their illness for progressing severity for a few days or would seek care at a local pharmacy or clinic before taking the time and effort to attend a busy hospital OPD. In addition to this fact, the study required a documented fever in the outpatient department at the time of enrollment. Thus, we expected greater illness severity and higher rate of hospitalization among these selected patients. We cannot explain why only 1% of study participants required admission and none required ICU care or died. Few

required follow up care even. This does suggest that the true morbidity of febrile illnesses including enteric fever, rickettsiosis, dengue fever and leptospirosis may have a wide range with most patients having a self-limited illness. Additional studies with similar designs and larger sample sizes may be needed to truly understand the burden of illness associated with various infectious pathogens and determine on which of these pathogens to focus public health mitigation and prevention efforts.

## Limitations

This analysis is subject to several limitations. Even after testing for multiple pathogens, the causes of 67% of AFI cases remained unknown. We did not conduct tests for common respiratory viruses, such as influenza, so we were not able to describe the contribution of influenza and other respiratory infections to AFI. However, there is a strong hospital-based surveillance system for influenza and other respiratory viruses in place in Bangladesh [75–77] which can supplement this gap. Moreover, unexpected premature suspension of patient enrolment due to the detection of COVID-19 cases in Bangladesh in March 2020 resulted in a reduced number of cases screened and led to lack of paired sera for some testing, which may have limited further detections. Since our sample size is small for detected pathogens, it is more likely to find insignificant variables that could be significant when the sample size increases. We have tried to overcome this limitation using penalized logistic regression technique during data analysis. Another limitation was that this surveillance cannot describe true seasonality because it had a duration of less than one calendar year. However, we reported findings of 11 months and will restart AFI surveillance for multiple years to better describe seasonality in the future. After the emergence of COVID-19 pandemic, all future AFI studies should consider testing SARS-CoV-2.

## Conclusion

In Bangladesh, this surveillance contributes useful diagnostic and epidemiologic exploration of acute febrile illness in patients aged 2 years and older. The use of highly sensitive diagnostics and state-of-the-art laboratory techniques to better characterize the pathogens responsible for acute fever in this region is the analysis' novelty and strength. The pathogens found in febrile patients and their predictors may lead to accurate clinical diagnosis and rapid treatment as well as enhanced global health security by adopting appropriate control and prevention measures.

## Supporting information

**S1 Table. List of pathogens and tests used for AFI surveillance, Bangladesh.**
(DOCX)

## Acknowledgments

icddr,b acknowledges with gratitude the commitment of the US CDC to its research efforts. icddr,b is also grateful to the Governments of Bangladesh, Canada, Sweden, and the UK for providing core/unrestricted support. We greatly acknowledge Dr. Firdausi Qadri, Dr. Tahmeed Ahmed, the directors, clinicians, teachers and staff of the surveillance hospitals for their support to conduct this study and are grateful to our study participants for their contributions.

## Author Contributions

**Conceptualization:** Pritimoy Das, M. Ziaur Rahman, Mahmudur Rahman, Mohammod Jobayer Chisti, Daniel W. Martin, Michael Friedman.

**Data curation:** Pritimoy Das, Anik Palit, Daniel W. Martin.

**Formal analysis:** Pritimoy Das, Daniel W. Martin, Angella Sandra Namwase, Cecilia Y. Kato, Jeri Stewart-Juba, Marah Condit, Ida H. Chung, Renee Galloway, Adam L. Cohen.

**Funding acquisition:** Pritimoy Das, Michael Friedman.

**Investigation:** Pritimoy Das, M. Ziaur Rahman, Mohammod Jobayer Chisti, Michael Friedman.

**Methodology:** Pritimoy Das, M. Ziaur Rahman, Mahmudur Rahman, Mohammod Jobayer Chisti, Daniel W. Martin, Mahabub Ul Anwar, Pawan Angra, Michael Friedman.

**Project administration:** Pritimoy Das, Mahmudur Rahman, Mohammod Jobayer Chisti, Mahabub Ul Anwar, Michael Friedman.

**Resources:** Pritimoy Das, Cecilia Y. Kato, Michael Friedman.

**Supervision:** Pritimoy Das, Sayera Banu, Mahmudur Rahman, Mohammod Jobayer Chisti, Daniel W. Martin, Mahabub Ul Anwar, Michael Friedman, Adam L. Cohen.

**Visualization:** Pritimoy Das, Fahmida Chowdhury, Adam L. Cohen.

**Writing – original draft:** Pritimoy Das.

**Writing – review & editing:** Pritimoy Das, M. Ziaur Rahman, Sayera Banu, Mahmudur Rahman, Mohammod Jobayer Chisti, Fahmida Chowdhury, Zubair Akhtar, Anik Palit, Daniel W. Martin, Mahabub Ul Anwar, Angella Sandra Namwase, Pawan Angra, Cecilia Y. Kato, Carmen J. Ramos, Joseph Singleton, Jeri Stewart-Juba, Nikita Patel, Marah Condit, Ida H. Chung, Renee Galloway, Michael Friedman, Adam L. Cohen.

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
