## [Decision Letter · Decision Letter 0]

13 Jul 2022

PONE-D-22-07249Acute Febrile Illness Among Outpatients Seeking Health Care in Bangladeshi Hospitals Prior to the COVID-19 Pandemic.PLOS ONE

Dear Dr. Das,

Thank you for submitting your manuscript to PLOS ONE. After careful consideration, we feel that it has merit but does not fully meet PLOS ONE’s publication criteria as it currently stands. Therefore, we invite you to submit a revised version of the manuscript that addresses the points raised during the review process.

ACADEMIC EDITOR: Please revise the paper according to the comments of reviewers and resubmit for consideration.

We look forward to receiving your revised manuscript.

Kind regards,

Aneesh Basheer

Academic Editor

PLOS ONE

“This research protocol was funded by the Centers for Disease Control and Prevention (USA) under the cooperative agreement no. GH002259.”

“This research protocol was funded by the Centers for Disease Control and Prevention (USA) under the cooperative agreement no. GH002259.”

“This research protocol was funded by the Centers for Disease Control and Prevention (USA) under the cooperative agreement no. GH002259.”

7. Your ethics statement should only appear in the Methods section of your manuscript. If your ethics statement is written in any section besides the Methods, please delete it from any other section.

8. We note that [Figure 1] in your submission contain [map/satellite] images which may be copyrighted. All PLOS content is published under the Creative Commons Attribution License (CC BY 4.0), which means that the manuscript, images, and Supporting Information files will be freely available online, and any third party is permitted to access, download, copy, distribute, and use these materials in any way, even commercially, with proper attribution. For these reasons, we cannot publish previously copyrighted maps or satellite images created using proprietary data, such as Google software (Google Maps, Street View, and Earth). For more information, see our copyright guidelines: http://journals.plos.org/plosone/s/licenses-and-copyright.

Natural Earth (public domain): http://www.naturalearthdata.com/.

Additional Editor Comments:

Please modify the manuscript according to comments by reviewers.

Reviewers' comments:

Reviewer's Responses to Questions

**Comments to the Author**

1. Is the manuscript technically sound, and do the data support the conclusions?

Reviewer #1: Partly

Reviewer #2: Partly

2. Has the statistical analysis been performed appropriately and rigorously? 

Reviewer #1: Yes

Reviewer #2: Yes

3. Have the authors made all data underlying the findings in their manuscript fully available?

Reviewer #1: Yes

Reviewer #2: Yes

4. Is the manuscript presented in an intelligible fashion and written in standard English?

Reviewer #1: Yes

Reviewer #2: Yes

5. Review Comments to the Author

Reviewer #1: Many thanks for the opportunity to review the manuscript. I think this is an important piece of research which will have impact on the care of patients with AFI. The study has recruited well from a wide geographical area and the multiple testing strategies were robust. There are a few parts of the manuscript which could require attention and would improve the overall publication.

1. Was there really no senior author based in Bangladesh?

2. In the introduction i think more emphasis should be given to the fact that AFI is often undifferentiated making clinical diagnosis more difficult, hence the need for broad diagnostic testing. I would reference the work of the Fiebre study in this regard.

3. In the writing, there are examples of a non-academic style, for example 'very young kid' and 'faraway hospitals'. These need more carefully defining and clarifying.

4. The age profile of enteric fever cases in your study is problematic - the SEAP study which is referenced and the STRATAA study which should also be referenced (Lancet Global Health, 2021), showed that in Bangladesh the burden of disease for enteric fever is in young and school aged children. These were both much larger, incidence-based studies and therefore provide more robust evidence. I think you should remove the sentence on age implications of typhoid conjugate vaccine using data from this manuscript as i think these other studies are show something quite different.

5. Can you compare the differing rates of detection through the year with the monsoon rains.

6. The reason for an increase in males with enteric fever could be due to differing healthcare utilisation practices rather than the conclusion you have drawn and should be mentioned.

7. The direct efficacy of TCV from the Qadri et al study was 81%. I do not think this qualifies as 'low' as stated in the manuscript. The 57% figure that is quoted was for overall protection of the vaccine including non-vaccinated individuals, i think this paragraph needs reworking.

Reviewer #2: 1. what is the reason to choose 100.4 as cut off for temperature in the inclusion criteria

2. Did patients diagnosed to have UTI have symptoms of UTI? If so shoulsd that not be an exclusion criteria since this a study clinically undiagnosed fevers.

3. Did patients with Derngue and leptospirosis have any clinical clues pointing to the diagnosios.

4. Will the outcome of this study change the current practice of empirical treatment of acute febrile illness

6. PLOS authors have the option to publish the peer review history of their article (what does this mean?). If published, this will include your full peer review and any attached files.

Reviewer #1: **Yes: **James Meiring

Reviewer #2: No

---

## [Author Response · Author response to Decision Letter 0]

5 Aug 2022

Response to the academic editor’s comments

Comment 1. Please ensure that your manuscript meets PLOS ONE's style requirements, including those for file naming. The PLOS ONE style templates can be found at

Response: Thanks for the valuable suggestion. We have updated this manuscript following the PLOS ONE's style requirements and the current version reflects this.

“This research protocol was funded by the Centers for Disease Control and Prevention (USA) under the cooperative agreement no. GH002259.”

Response: We have included the updated statement as follows: “The funders had no role in study design, data collection and analysis, decision to publish, or preparation of the manuscript.”

“This research protocol was funded by the Centers for Disease Control and Prevention (USA) under the cooperative agreement no. GH002259.”

“This research protocol was funded by the Centers for Disease Control and Prevention (USA) under the cooperative agreement no. GH002259.”

Response: We have removed funding-related text from the manuscript under the acknowledgement section and Funding section as suggested. Please include the Funding Statement in the online submission form for this work as follows: “This research protocol was funded by the Centers for Disease Control and Prevention (USA) under the cooperative agreement no. GH002259. The funder had no role in study design, data collection and analysis, decision to publish, or preparation of the manuscript.”

Response: Thanks for the comment. We have updated our Data Availability statement and included how to access data as (Line number 465-472) “Data cannot be made publicly available because this Human Subject Research dataset contains potentially sensitive information and hence, are confidential in ethical perspective. icddr,b recognizes the public health, social and intellectual value of providing access to its knowledge data. Data will be provided to interested researchers (Recipients) for upon approval of a Data Licensing Application & Agreement (DLAA) by the icddr,b Data Centre Committee (DCC). Request for icddr,b research data should be addressed to Ms. Armana Ahmed, Head, Research Administration at aahmed@icddrb.org.” 

Response: Yes, we have updated the “Data Availability statement” including all the relevant information suggested in the previous comment.

Response: Please accept my sincere apologies. It came from an earlier draft of the manuscript and by mistake, was not deleted. I have removed “data not shown” from the updated manuscript. 

7. Your ethics statement should only appear in the Methods section of your manuscript. If your ethics statement is written in any section besides the Methods, please delete it from any other section.

Response: Thanks. We have deleted ethics statement from any other part of the updated manuscript except the methods section.

8. We note that [Figure 1] in your submission contain [map/satellite] images which may be copyrighted. All PLOS content is published under the Creative Commons Attribution License (CC BY 4.0), which means that the manuscript, images, and Supporting Information files will be freely available online, and any third party is permitted to access, download, copy, distribute, and use these materials in any way, even commercially, with proper attribution. For these reasons, we cannot publish previously copyrighted maps or satellite images created using proprietary data, such as Google software (Google Maps, Street View, and Earth). For more information, see our copyright guidelines: http://journals.plos.org/plosone/s/licenses-and-copyright.

Natural Earth (public domain): http://www.naturalearthdata.com/.

Response: Thanks for notifying this and sharing all those resources. I have removed the map from the figure 1 and inserted new map of my own creation. This now ensures that there is no use of any copy-right protected map in figure 1. Please find the updated ‘Figure 1’ with the revised submission. 

Response: To our knowledge, our reference list it is complete and correct.

Response to Reviewer #1 comments

Reviewer #1: Many thanks for the opportunity to review the manuscript. I think this is an important piece of research which will have impact on the care of patients with AFI. The study has recruited well from a wide geographical area and the multiple testing strategies were robust. There are a few parts of the manuscript which could require attention and would improve the overall publication.

1. Was there really no senior author based in Bangladesh?

Response: Thanks for the important comments and valuable review feedback. We accept that Bangladesh has very renown researchers in the field of public health (some of them are acknowledged in the ‘acknowledgement’ section). However, the authorship of this AFI manuscript was based upon the involvement and contribution in this particular acute febrile illness project, which was led by a group of icddr,b and CDC, USA researchers. Based on their scholarly contribution in this study, we have developed the author-line. 

2. In the introduction i think more emphasis should be given to the fact that AFI is often undifferentiated making clinical diagnosis more difficult, hence the need for broad diagnostic testing. I would reference the work of the Fiebre study in this regard.

Response: We have updated some sentences in the introduction to reflect the difficulty in AFI diagnosis based on the important suggestion and also added a few sentences.

Line 70-76: “Due to the lack of rapid diagnostic capacity, patients suffering from AFI, particularly early in the clinical course when no symptoms can distinguish different aetiologies, pose challenges to their physicians at out-patient departments of any hospital…. Clinically, the different causes of acute febrile illnesses may be indistinguishable, and the choice of empiric antibiotics is determined by the etiologic profile, which varies by time, place, and personal factors.” 

Line 76-81: “To fill the gaps, there is a need for a broad diagnostic testing approach. Recently Hopkins et al. (2020) has taken an initiative through The Febrile Illness Evaluation in a Broad Range of Endemicities (FIEBRE) study to help address these information gaps. FIEBRE was intended to explore AFI in paediatric and adult outpatients and inpatients, using standardised clinical, reference laboratory and social science protocols, in low-resource regions from five sites in sub-Saharan Africa and South-eastern and Southern Asia.”

3. In the writing, there are examples of a non-academic style, for example 'very young kid' and 'faraway hospitals'. These need more carefully defining and clarifying.

Response: We have corrected the words in the manuscript as follows-

Line 208-210: “For example, a 2-year-old female child was unable to provide urine in outpatient settings, or a delayed (>24 hour) arrival of urine from Rajshahi hospital, a facility far from the Dhaka icddr,b laboratory)”.

4. The age profile of enteric fever cases in your study is problematic - the SEAP study which is referenced and the STRATAA study which should also be referenced (Lancet Global Health, 2021), showed that in Bangladesh the burden of disease for enteric fever is in young and school aged children. These were both much larger, incidence-based studies and therefore provide more robust evidence. I think you should remove the sentence on age implications of typhoid conjugate vaccine using data from this manuscript as i think these other studies are show something quite different.

Response: We have updated the manuscript including citing from the STRATAA study. 

Line 358-361: “A multi-country population-based study found that the seroincidence of Salmonella enterica serotype Typhi exposure in Bangladesh was highest in 15-29 years age group and individuals aged 5 to 9 years had the highest adjusted incidence of typhoid fever.” We have also removed the conflicting age implication sentence as mentioned in the comment. 

5. Can you compare the differing rates of detection through the year with the monsoon rains.

Response: It would be good, but we do not have sufficient data for such analysis. Sorry for that.

6. The reason for an increase in males with enteric fever could be due to differing healthcare utilisation practices rather than the conclusion you have drawn and should be mentioned.

Response: We included in line number 364: “In addition, the reason for an increase in males with enteric fever could be due to differing healthcare utilization practices.”

7. The direct efficacy of TCV from the Qadri et al study was 81%. I do not think this qualifies as 'low' as stated in the manuscript. The 57% figure that is quoted was for overall protection of the vaccine including non-vaccinated individuals, i think this paragraph needs reworking.

Response: Thanks for noticing this unintentional mistake and the interpretation of that result. We have updated the manuscript in line number 372 as “Vi-tetanus toxoid conjugate vaccine efficacy against typhoid fever was ranging from 80% to 88% for vaccinated children in urban Dhaka.”

Response to Reviewer #2 comments

Reviewer #2: 

1. what is the reason to choose 100.4 as cut off for temperature in the inclusion criteria

Response: Thanks for the valuable comments. Normal body temperature is usually 37 degrees Celsius (37°C) or 98.6 degrees Fahrenheit (98.6°F). Most of the clinical settings and the majority of febrile illness studies published in peer-reviewed journals, a fever is defined as a body temperature of 38°C (100.4°F) or higher. World Health Organization uses fever cut-off at 100.4°F core temperature in their IMCI guideline for integrated management of childhood illness. Directorate General of Health Services, Ministry of Health & Family Welfare, Bangladesh published IMCI Managers Toolkit which also considers 38°C or 100.4°F as fever (page 19, IMCI managers toolkit 2019, Government of the people’s republic of Bangladesh). There is, of course, some variation in some other literatures. To ensure that our study results are comparable with most of the publications, we selected 100.4°F as cut off for temperature in our study. 

2. Did patients diagnosed to have UTI have symptoms of UTI? If so should that not be an exclusion criteria since this a study clinically undiagnosed fevers.

Response: Thanks for the concern. We have excluded patients with symptoms of a focused infections (Line number 109). Since we were interested in the cause of fever without an obvious source; therefore, cases with obvious signs or symptoms of UTI were excluded. The study physician(s) did not enrol a patient with UTI diagnosis based on the presenting sign-symptoms during the initial assessment at the outpatient department. 

3. Did patients with Dengue and leptospirosis have any clinical clues pointing to the diagnosis.

Response: The current manuscript is aetiology-focused, and hence, does not have the scope of pointing detailed clinical diagnosis which requires specialized analysis. We are under the development of another manuscript which will analyse clinical clues pointing the diagnosis using a 5-fold cross-validated chi-squared automatic interaction detection (CHAID) algorithm. We hope that we will be able to answer your important question. 

4. Will the outcome of this study change the current practice of empirical treatment of acute febrile illness

Response: Our study findings are based on the use of highly sensitive diagnostics and state-of-the-art laboratory techniques. We have disseminated our preliminary finding among the clinicians, directors of the hospitals and administrative authorities of the Directorate General of Health Services, Bangladesh. We have received positive feedback from them. Globally, it is beyond our scope to predict that our study would impact the existing practice of empirical treatment of acute febrile illness; nonetheless, we are confident that our study offered fascinating results and contributed to our understanding of acute febrile illness.

---

## [Editor Report · Decision Letter 1]

18 Aug 2022

Acute febrile Illness among outpatients seeking health care in Bangladeshi hospitals prior to the COVID-19 pandemic.

PONE-D-22-07249R1

Dear Dr. Das,

We’re pleased to inform you that your manuscript has been judged scientifically suitable for publication and will be formally accepted for publication once it meets all outstanding technical requirements.

Kind regards,

Aneesh Basheer

Academic Editor

PLOS ONE
---

## [Editor Report · Acceptance letter]

22 Aug 2022

PONE-D-22-07249R1 

Acute febrile illness among outpatients seeking health care in Bangladeshi hospitals prior to the COVID-19 pandemic 

Dear Dr. Das:

I'm pleased to inform you that your manuscript has been deemed suitable for publication in PLOS ONE. Congratulations! Your manuscript is now with our production department. 

Kind regards, 

on behalf of

Dr. Aneesh Basheer 

Academic Editor

PLOS ONE